# Carpal Tunnel Syndrome and Ulnar Nerve Entrapment Are Associated with Impaired Psychological Health in Adults as Appraised by Their Increased Use of Psychotropic Medication

**DOI:** 10.3390/jcm11133871

**Published:** 2022-07-04

**Authors:** Lars B. Dahlin, Raquel Perez, Erika Nyman, Malin Zimmerman, Juan Merlo

**Affiliations:** 1Department of Translational Medicine—Hand Surgery, Lund University, 20502 Malmö, Sweden; malin.zimmerman@med.lu.se; 2Department of Hand Surgery, Skåne University Hospital, 20502 Malmö, Sweden; 3Department of Biomedical and Clinical Sciences, Linköping University, 58183 Linköping, Sweden; erika.nyman@liu.se; 4Unit for Social Epidemiology, Department of Clinical Sciences (Malmö), Faculty of Medicine, Lund University, 20502 Malmö, Sweden; raquel.perez@med.lu.se (R.P.); juan.merlo@med.lu.se (J.M.); 5Department of Hand Surgery, Plastic Surgery and Burns, Linköping University Hospital, 58183 Linköping, Sweden; 6Center for Primary Health Research, Region Skåne, 20502 Malmö, Sweden

**Keywords:** nerve compression, carpal tunnel syndrome, carpal tunnel surgery, ulnar nerve entrapment, cubital tunnel syndrome, psychotropic drugs, psychological health, socioeconomical factors, national quality register

## Abstract

We aimed to study psychological health, as approximated by the use of psychotropic drugs, in a population diagnosed and surgically treated for carpal tunnel syndrome (CTS) or ulnar nerve entrapment (UNE), or both, also considering the demographic and socioeconomic factors of the individuals. Linking data from five large national registers, use of psychotropics (at least one dispensation during the first year after the surgery or the baseline date) was examined in around 5.8 million people 25–80 years old residing in Sweden 2010. Among these individuals, 9728 (0.17%), 890 (0.02%) and 149 (0.00%) were identified as diagnosed and surgically treated for CTS, UNE, or both, respectively. As much as 28%, 34% and 36% in each group, respectively, used psychotropic drugs, compared with 19% in the general population. Regression analyses showed a general higher risk for use of psychotropics related to these nerve compression disorders, to higher age, being a woman, and having low income or low occupational qualification level. Individuals born outside of Sweden had a lower risk. We conclude that surgically treated individuals with a nerve compression disorder have an increased risk of impaired psychological health. Caregivers should be aware of the risk and provide necessary attention.

## 1. Introduction

The two common nerve compression disorders, carpal tunnel syndrome (CTS) and ulnar nerve compression at the elbow or wrist (both here defined and abbreviated as UNE), induce symptoms and disability, which may severely affect the individuals’ life, particularly if pain is a major clinical component [1,2,3,4]. Nerve compression disorders, such as CTS and UNE, have an incidence of 105–197 and 26–36 per 100,000 person-years, respectively, of whom around 62% and 45%, respectively, are surgically treated [5,6]. Socio-economic factors have been discussed in the context of the risk of being diagnosed and treated for CTS in particular [7,8,9,10,11,12], but has been less highlighted for UNE [1,13,14]. More importantly, the recognition and addressing of psychological health early is crucial in decision-making before performing surgery for CTS and UNE [15], and also an important factor to consider during rehabilitation [16], and it may influence return to work after carpal tunnel surgery [17]. The use of psychotropics (i.e., psycholeptics, antidepressants, and psycholeptics and psychoanaleptics in combination) can be considered as a proxy or indicator for impaired psychological health, especially in highly accessible health care systems such as the Swedish system, and can be used to evaluate if a condition is associated with impaired psychological health [18]. Using psychotropic drugs as an indicator of psychological health, it has previously been found that children born with a brachial plexus birth injury have an increased risk of suffering poor psychological health during adolescence [18]. In addition, children born with an orofacial cleft have an increased risk of psychotropic drug use compared to children born only with a cleft lip or cleft palate [19]. Thus, one may hypothesize that even nerve compression disorders, such as CTS and UNE, or the combination of both disorders [20], may have an impact on the use of psychotropic drugs among surgically treated individuals over and above the discussed socioeconomical factors related to nerve compression disorders.

Our aim was to study the risk of impaired psychological health, as approximated by the use of psychotropic drugs, in relation to the existence of diagnoses and surgical treatments for the nerve compression disorders CTS and UNE alone or in combination. When doing so, we also considered the demographical and socioeconomic characteristics of the included surgically treated individuals.

## 2. Population & Methods

### 2.1. Databases

The present record linkage study joined data from several registers with individual level information covering the whole population of Sweden. We used data from the registers of the Total Swedish Population (TPR) and the Longitudinal Integration Database for Health Insurance and Labor Market Studies (LISA), administered by Statistics Sweden (www.scb.se/en/, accessed on 1 January 2021), as well as from the National Patient Register (NPR), the Cause of Death Register (CDR) and the Swedish Prescribed Drug Register (SPDR), administered by the National Board of Health and Welfare (www.socialstyrelsen.se/en/, accessed on 1 January 2021). After revision and consent by their own data safety committees and initial approval by the Regional Ethical Committee in South Sweden (#: 2014-856), the Swedish authorities anonymized the registers and delivered them to us. The record linkage was performed by us using a unique anonymized personal identification number.

The SPDR contains information about all drug dispensations in the Swedish pharmacies, except from stockpiles in nursing homes and hospital wards, coded according to the Anatomical Therapeutic Chemical (ATC) classification system, while the NPR codes discharge diagnoses from hospital and outpatient clinics according to the International Classification of Diseases and Causes of Death, 10th version (ICD-10). The NPR also records and codes clinical and surgical procedures according to the Swedish Classification of Care Procedures (SCCP). The TPR and the LISA database provide demographic and socioeconomic information.

Our research database consisted of the total Swedish population of 2010 (i.e., residing in Sweden 31 December 2010). From the approximately 9.4 million people, we excluded those who died (*n* = 95,618) or emigrated (*n* = 49,939) during the one year follow up, individuals residing less than five years in the country (*n* = 423,414), those whose sex was not registered (*n* = 335) and people without information on country of birth (COB) (*n* = 60,564). We also excluded people below the age of 25 years and above the age of 80 (*n* = 3,027,306). In addition, we excluded those with previous CTS-UNE operation (*n* = 12,250). The final sample consisted of around 5.8 million people (Figure 1).

### 2.2. Assessment of Variables

We defined nerve compression disorders according to the ICD-10 and SCCP codes simultaneously registered at the hospital discharge as *surgery for carpal tunnel syndrome* (CTS) (ICD-10: G56.0 and SCCP: ACC51) and *surgery for ulnar nerve compression at the elbow or wrist* (both here defined and abbreviated as UNE) (ICD-10: G56.2 and SCCP: ACC53). In the analyses, we distinguished between CTS, UNE or both CTS and UNE if the hospital discharge simultaneously presented both diagnoses and surgical procedures.

We assigned an individual baseline date to every individual, defined by the date of the first CTS/UNE diagnosis in 2011, or 1 January 2011 if the individual did not have any CTS/UNE diagnosis. Thereafter, we followed everyone for one year from the baseline date (the follow-up could extend until 31 December 2012) in order to ascertain their use of psychotropics defined as at least one dispensation (ATC code) of Psycholeptics (N05), Antidepressants (N06A) or Psycholeptics and Psychoanaleptics in combination (N06C). We considered use of psychotropics as a proxy for impaired psychological health as discussed elsewhere [18].

Previous psychotropic drug use defined at any dispensation of Psycholeptics (N05), Antidepressant (N06A) or Psycholeptics and Psychoanaleptics in combination (N06C) five years before the baseline.

Age was arbitrarily classed into five wide categories, i.e., 25–34 (reference), 35–44, 45–54, 55–64 and 65–80 year-olds. In age-stratified regression analyses, we included continuous age as a quadratic function. We used essentially ten-year categories for descriptive purposes in Figure 2 (25–34, 35–44, 45–54, 55–64 and 65–80 year-olds). Sex was coded as male (reference) or female according to the register. We categorized the individuals according to their COB as born in Sweden (reference) or not (i.e., immigrant). We obtained information on individualized disposable family income for the years 2000, 2005 and 2010 to compute a cumulative measure that considers the size of the household and the consumption weight of the individuals according to Statistics Sweden. For each of the three years, income levels were categorized into 25 groups (1–25) by quantiles using the complete Swedish population. These groups from the respective three years were summed up, so that everyone received a value between 3 (always in the lowest income group) and 75 (always in the highest income group). We categorized this cumulative income into three groups by tertiles [low, medium or high (reference) income]. Individuals with missing values on income during 2000 or 2005 were assigned the values for the year 2010. No individuals had missing income data for 2010.

Occupational qualification level was categorized into five skill groups, which reflect the type of working task and their complexity (low, middle-low, middle-high, high, and missing) according to the Swedish Standard Classification of Occupations 2012 [21]. The major groups in SSYK2012 are associated with the following skill levels: managers, commissioned officers and occupation requiring advanced level of higher education were classified as high level; occupation managers in service industries, occupations requiring higher education qualifications or equivalent and non-commissioned officers were classified as middle-high; administration and customer service clerks, service, care and shop sales workers, agricultural, horticultural, forestry and fishery workers, building and manufacturing and transport workers and manufacturing and transport workers were classified as middle–low; elementary occupation was classified as low level. If no information was available [553,865 individuals (9.61%)], the case was classified as missing. The distribution of age in cases with no information in occupation was 15.9% between 25 and 34 years old, 13.2% between 35 and 44, 17.4% between 45 and 54, 24.1% between 55 and 64 and 29.5% 65 or more years old.

### 2.3. Statistical Analyses

We performed age-stratified analyses to calculate the absolute risk (AR), the absolute risk difference (ARD) and 95% confidence intervals of use of psychotropics in relation to the existence of the nerve compression disorders.

Since prevalence of the outcomes was relatively high, we measured the relative associations between the explanatory variables and use of psychoactive drugs by prevalence ratios (PRs) rather than by odds ratios [22]. For this purpose, we applied Cox proportional hazards regression models with a constant follow-up time equal to 1. We developed two regression models. Model 1 included only the nerve compression disorders and model 2 added socioeconomic and demographic variables (i.e., age, sex, income, country of birth and occupational qualification level).

We estimated the discriminatory accuracy (DA) for each model by calculating the area under the receiver operating characteristic curve (AUC) and its 95% confidence intervals (CI). The value of the AUC ranges from 0.5 to 1, with 1 representing perfect discrimination and 0.5 indicating no predictive accuracy [23]. Using the criteria proposed by Hosmer and Lemeshow [24], we classified DA as absent or very weak (AUC = 0.5–0.6), poor (AUC >0.6–≤ 0.7), acceptable (AUC > 0.7–≤ 0.8) or excellent (AUC > 0.8–0.90) and outstanding (AUC > 0–90).

## 3. Results

### 3.1. Demographic and Socioeconomic Characteristics of the Population

The characteristics of the surgically treated individuals with the actual nerve compression disorders, CTS, UNE or both, residing in Sweden by 2010 and included in the study, are presented in Table 1. Among the 5.8 million individuals 25–80 years old, 9728 (0.17%) individuals had CTS, 890 (0.02%) individuals had UNE and in addition 149 (0.00%) individuals were surgically treated for a diagnosis of both CTS and UNE.

The age distribution of the nerve compression disorders showed a slight increase of CTS with age and with a peak between 45 and 54 years of age (Table 1). Overall, there were more women than men among individuals with CTS and among individuals with both CTS and UNE, while the sex distribution was equal among the individuals with UNE (Table 1). Compared with the group with no nerve compression disorders, there were essentially small differences in the distribution of income, but high-income individuals were underrepresented and low-income overrepresented in the groups with the three different nerve compression disorders (Table 1). The number of immigrants with diagnosed and surgically treated nerve compression disorders was lower, but proportionally rather similar across the nerve compression disorder categories. The occupational qualification levels showed a difference with higher proportions of individuals with low and middle low qualification levels in individuals with nerve compression disorders. In accordance, the proportions were lower among individuals with middle-high and high qualifications. Missing information on occupational qualification level was rather similarly distributed across the categories of nerve compression disorders (Table 1).

### 3.2. Use of Psychotropic Medication

Of the adults diagnosed with CTS, UNE and both disorders, 28%, 34% and 36%, respectively, used psychotropic drugs, while this proportion was only 19% in the adult general population (Table 1). The use of psychotropic drugs across the age groups, divided by sex and the three different nerve compression disorders are presented in Figure 2. Overall, women had a higher use of psychotropic drugs than men and there were higher proportions of use of psychotropic drugs among surgically treated individuals with nerve compression disorders than among individuals without such disorders across all age categories except for men older than 54 years.

We used two different regression models to analyze the use of psychotropic drugs (Table 2). In the first model, including only the nerve compression disorders, there was an increased risk for the use of psychotropic drugs compared to the individuals without nerve compression disorders. In relative terms, this increased risk was lower in individuals with CTS (Prevalence Ratio; PR = 1.50) than in individuals with UNE (PR = 1.83) or with the combination of both the nerve compression disorders (PR = 1.90). In the second model including age, sex, income, occupational qualification level, COB and previous psychotropic use. The adjusted PRs of the nerve compression disorders were reduced but remained conclusively high. Higher age, being a woman, having a low income and a low occupational qualification level increased the risk for use of psychotropic drugs (Table 2). Being born outside Sweden showed a slight, but non-significant, lower risk for use of psychotropic drugs. As expected, previous use of psychotropic medication was strongly associated with psychotropic use after the operation. (PR = 15.59).

Table 3 shows age stratified unadjusted AR and ARD of psychotropic medication use during 2010 in relation to CTS, UNE, or both in the 5,751,152 individuals aged 25–80 years residing in Sweden by 2010. The unadjusted AR of psychotropic medication use was higher in all the categories of nerve compression disorders than in the general population without such disorders, except for in the oldest age group with both diagnoses. Overall, the ARD was systematically higher in the three categories of the surgically treated individuals with the nerve compression disorder than in those without such a diagnosis (Table 3). However, the 95% CI showed a large uncertainty in elderly individuals with UNE or both CTS-UNE categories.

## 4. Discussion

The present age stratified analysis of 5.8 million individuals, 25–80 years and residing at least five years in Sweden by 2010, provides observational evidence of the existence of impaired psychological health after surgery for the two common nerve compression disorders, CTS and UNE, or for a combination of the two conditions. The use of psychotropic drugs can be considered as an indicator for impaired psychological health, and this app-roach has been previously applied for other injuries and disorders, such as brachial plexus birth injury [18] as well as orofacial cleft [19]. In the unadjusted analysis, both absolute and relative risk of psychotropic medication use was much higher in individuals that were diagnosed and surgically treated for CTS and UNE or both conditions. The combination of these two nerve compression disorders simultaneously is not common from a population perspective, but rather common from a clinical perspective as to why the individuals appearing with such a combination were also included in the present study [2]. In the regression models, we found that this increased risk of psychotropic drug use in surgically treated individuals with CTS, UNE and the combination of CTS and UNE remained when adjusted for age, sex, income, country of birth and occupational qualifications with PRs of about 1.50. In the same models, PRs for the use of psychotropic drugs were also increased by a higher age, being a woman, having low income and a low occupational qualification level. Regarding the relevance of these factors, one may consider that socioeconomic status can be a confounder from the perspective of nerve compression disorders and impaired psychological health. However, our adjusted analyses (Table 2, model 2) indicated an independent effect of both the nerve compression disorders and the socioeconomic status. A positive association between anxiety, depression, and health-related quality of life with patient-reported symptom severity, but not for objectively derived severity, of CTS has been reported [25]. Some authors argue that electrophysiological testing, as a more objective assessment, should be performed before considering surgery in CTS due to the risk that poor mental health results in functional symptoms [26].

There are socioeconomic disparities among individuals surgically treated for CTS, where socially deprived patients are less likely to receive surgical treatment in an American setting [11]. Whether this is also the case in countries such as Sweden, where health care is financed by the government and equally available regardless of socioeconomic status, remains unknown. However, in our database most patients with a diagnosis received surgical treatment. Economic well-being seems to be crucial in CTS since a low economic well-being is related to higher comorbidity burden [12], such as diabetes in which nerves are more susceptible to nerve compression [27,28]. There is also evidence that both type of occupation as well as level of educational achievement are important for development of clinically relevant CTS [7]. Most of these socioeconomical factors are associated with more symptoms both before and after surgery for CTS, but do not affect the relative improvement [9]. Analysis of preoperative psychological mindsets, based on several questionnaires, seems to be important for predicting the outcome of surgery [29]. Return to work after surgery for CTS is also influenced by illness perception and mental health [17], which seems to be particularly relevant for treated women for CTS where depression predicts outcome [16]. In the present study, a low income as well as a low occupational qualification level were associated with an increased odds ratio for use of psychotropic drugs over and above other factors. Thus, the etiological factors, often multifactorial, behind nerve compression disorders, such as CTS and the relation to mental health, such as depression and socioeconomic status, are complex, but should be considered in clinical practice when treating individuals with nerve compression disorders.

Most studies concerning the importance of mental illness and socioeconomical factors for having or being treated for a nerve compression disorder have so far been focused on CTS and there have been few on UNE [13]. CTS and UNE, requiring surgery, are more common in socially deprived individuals and seem to occur at an earlier age [15], where the authors stressed that the relationship was strikingly similar for UNE and CTS. We found a higher risk for use of psychotropic drugs among the surgically treated individuals with CTS, UNE and the combination of UNE and CTS, although the size of the risk was different between the three groups. The high use of psychotropic drugs among middle aged women with UNE is an interesting observation. Despite some similarities between CTS and UNE as nerve compression disorders, the latter condition is different in many aspects. This includes a more unpredictable outcome of surgery in some individuals with UNE, where psychological health may be one crucial influencing factor and a risk for postoperative neuropathic pain. Thus, the indication for surgery for UNE should be very clear and sometimes strict. An interesting question is if patients with preoperative anxiety and/or depression benefit from a specific type of surgery, but data indicate that patients, surgically treated with a total knee arthroplasty, are improved regardless of their presurgical psychological status [30]. Preoperative information, where also assessment of the psychological and psychiatric status is considered before surgery, seems to be crucial [31] and anxiety and depression symptoms may also decrease after a total knee arthroplasty [32]. Finally, we did not find any positive association between country of birth and the diagnosis or treatment of the three disorders; in fact, model 2 of the regression analysis indicated a decreased use of psychotropic drugs in surgically treated individuals born outside of Sweden. However, it is known that there is an underutilization of health care services and especially psychiatric medication by migrants [33,34].

Our study has several strengths. It is based on large national registers that cover the whole Swedish population and all the relevant patients. The coding of the diagnoses and surgical procedures is also highly standardized and similar across the whole country. However, it might also have some limitations. For instance, one may argue that our study did not have data from primary health care, but we defined the exposed population as patients diagnosed and surgically treated for a nerve compression disorder, which is a clear and valid definition of such syndromes, excluding transient and possibly vague symptoms from the peripheral nervous system. We also excluded 423,414 individuals residing less than five years in Sweden. We did so because it is known that there is an underutilization of health care services by migrants, especially those residing for only a few years in the country [34]. In addition, we aimed to obtain a solid measure of socioeconomic position based on information about income during the last 10 years and to obtain information about previous use of psychotropic medication and relevant surgery.

We conclude, based on a large record linkage database of several national registers covering the entire Swedish population at the age of 25–<80 years, that surgically treated individuals suffering from a nerve compression disorder, such as CTS and particularly UNE and a combination of both disorders, have an increased risk of impaired psychological health as expressed by their high use of psychotropic medication, and this risk is independent of the socioeconomic characteristics of the patients. Caregivers involved in the treatment of individuals with nerve compression disorders should be aware of such an increased risk and be ready to provide the necessary attention to the individual.

## Figures and Tables

**Figure 1 jcm-11-03871-f001:**
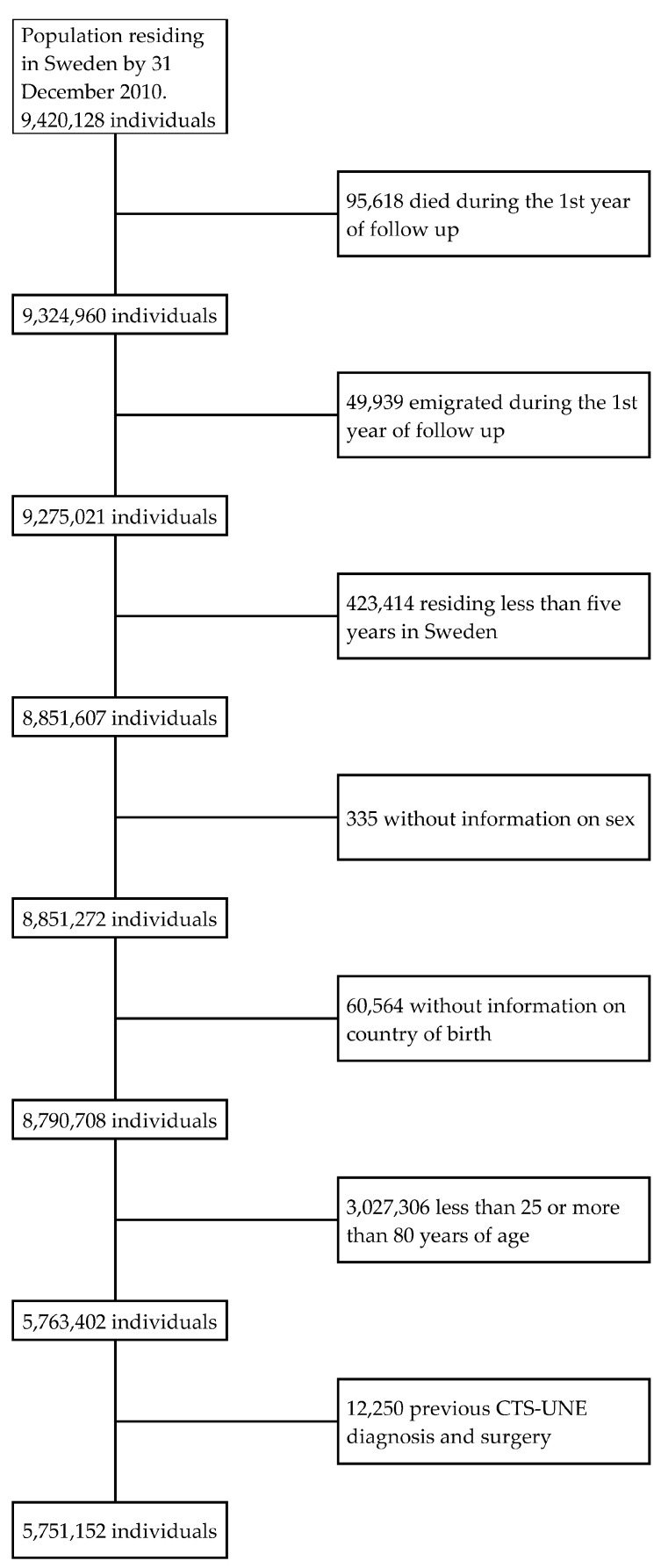
Flow chart showing the individuals excluded from the original 2010 Swedish population to obtain the final study sample.

**Figure 2 jcm-11-03871-f002:**
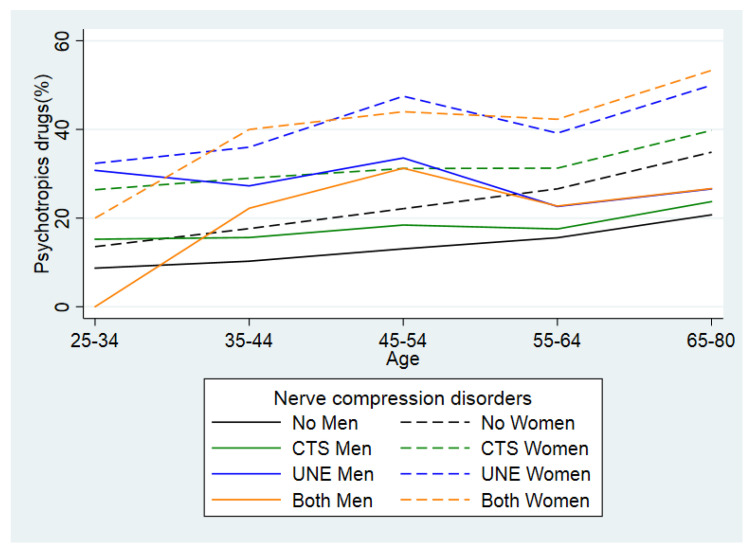
Age stratified percentage of use of psychotropic drugs in men and in women, residing in Sweden 2010, and suffering from carpal tunnel syndrome (CTS), ulnar nerve entrapment (UNE) and both disorders as well as in people without any nerve compression disorder.

**Table 1 jcm-11-03871-t001:** Characteristics of the 5,751,152 individuals residing in Sweden by 2010 and included in the study sample in relation to the existence of diagnosed and surgically treated carpal tunnel syndrome (CTS), ulnar nerve entrapment (UNE), or both by use of psychotropic drugs as well as demographical and socioeconomic factors. Values are number and percentages (%).

	Nerve Compression Disorders			
	None	CTS	UNE	Both
*Study sample*	5,740,385	9728	890	149
	(99.81)	(0.17)	(0.02)	(0.00)
*Psychotropic drugs*	1,072,677	2,732	305	53
	(18.69)	(28.08)	(34.27)	(35.57)
*Age (years)*				
25–34	997,168	803	73	6
	(17.37)	(8.25)	(8.20)	(4.03)
35–44	1,183,624	1689	188	24
	(20.62)	(17.36)	(21.12)	(16.11)
45–54	1,165,714	2522	260	41
	(20.31)	(25.93)	(29.21)	(27.52)
55–64	1,147,095	2205	234	41
	(19.98)	(22.67)	(26.29)	(32.21)
65–80	1,246,784	2509	135	30
	(21.72)	(25.79)	(15.17)	(20.13)
*Men*	2,865,465	3149	483	63
	(49.92)	(32.37)	(54.27)	(42.28)
*Income*				
Low	1,309,528	2520	236	31
	(22.81)	(25.90)	(26.52)	(20.81)
Middle	2,038,179	3646	351	57
	(35.51)	(37.48)	(39.44)	(38.26)
High	2,392,678	3562	303	61
	(41.68)	(36.62)	(34.04)	(40.94)
*Immigrant*	778,468	1150	118	24
	(13.56)	(11.82)	(13.26)	(16.11)
*Occupational qualification level*				
Low	329,645	804	68	15
	(5.74)	(8.26)	(7.64)	(10.07)
Middle-low	2,611,706	5469	525	73
	(45.50)	(56.22)	(58.99)	(48.99)
Middle-high	980,117	1300	94	28
	(17.07)	(13.36)	(10.56)	(18.79)
High	1,267,169	1326	115	18
	(22.07)	(13.63)	(12.92)	(12.08)
Missing	551,748	829	88	15
	(9.61)	(8.52)	(9.4889)	(10.07)
*Previous psychotropic use*	1,608,486	3993	440	80
	(28.02)	(41.05)	(49.44)	(53.69)

**Table 2 jcm-11-03871-t002:** Crude (Model 1) and adjusted (Model 2) for demographical and socioeconomic factors association between the existence of diagnosed and surgically treated carpal tunnel syndrome (CTS), ulnar nerve entrapment (UNE), or both and use of psychotropic drugs during the follow-up in the 5,751,152 individuals aged 25–80 residing in Sweden by 2010. Values are prevalence ratios (PR) and 95% confidence intervals (CI).

	Model 1	Model 2
	PR (95% CI)	PR (95% CI)
*Nerve compression disorders*		
None	Ref	Ref
CTS	1.50 (1.45–1.56)	1.06 (1.02–1.10)
UNE	1.83 (1.64–2.05)	1.16 (1.04–1.29)
Both	1.90 (1.45–2.49)	1.08 (0.83–1.42)
*Age (years)*		
25–34		Ref
35–44		1.12 (1.11–1.13)
45–54		1.20 (1.19–1.21)
55–64		1.30 (1.29–1.31)
65–80		1.49 (1.48–1.50)
*Men*		Ref
Women		1.15 (1.15–1.16)
*Income*		
Low		1.07 (1.07–1.08)
Middle		1.07 (1.06–1.07)
High		Ref
*Native*		Ref
Immigrants		0.89 (0.89–0.90)
*Occupational qualification level*		
Low		1.06 (1.05–1.07)
Middle-low		1.04 (1.03–1.04)
Middle-high		1.00 (1.00–1.01)
High		Ref
Missing		1.27 (1.26–1.42)
*Previous psychotropic drugs use*		
Yes		15.59 (15.50–15.67)
AUC	0.501	0.885

**Table 3 jcm-11-03871-t003:** Age stratified unadjusted absolute risk (AR), absolute risk difference (ARD) and 95% confidence intervals (CI) of psychotropic medication use during the follow-up in relation to the existence of diagnosed and surgically treated carpal tunnel syndrome (CTS), ulnar nerve entrapment (UNE), or both in the 5,751,152 individuals aged 25–80 years residing in Sweden by 2010.

	Nerve Compression Disorders											
Age (Years)	None			CTS			UNE			Both		
	N	AR	ARD	N	AR	ARD (95% CI)	N	AR	ARD (95% CI)	N	AR	ARD (95% CI)
25–34	997,168	11.09	Ref.	803	23.66	12.57 (9.63–15.51)	73	31.51	20.42 (9.76–31.07)	6	16.67	5.58 (−24.24–35.40)
35–44	1,183,624	13.90	Ref	1689	25.46	11.55 (9.48–13.36)	188	31.92	18.01 (11.35–24.68)	24	33.33	19.43 (0.57–38.29)
45–54	1,162,714	17.54	Ref	2522	27.87	10.33 (8.25–12.09)	260	40.00	22.46 (16.50–28.41)	41	39.02	21.48 (6.55–34.42)
55–64	1,147,095	21.10	Ref	2205	26.49	5.39 (3.54–7.23)	234	29.49	8.39 (2.55–14.23)	48	33.33	12.24 (−1.10–25.57)
65–80	1,246,784	28.15	Ref	2509	32.88	4.73 (2.89–6.57)	135	36.30	8.15 (0.03–16.26)	30	40.01	11.85 (−5.68–29.38)

## Data Availability

Relevant data are included within the paper. The complete and detailed individual data of all subjects cannot be publicly available for ethical and/or legal reasons due to compromising patient privacy. The Regional and National Ethical Committee have imposed these restrictions. Data requests may be sent to The National Ethical Committee via the homepage of Etikprövningsmyndigheten in Sweden (etikprovningsmyndigheten.se, accessed on 1 June 2022). The database we analyzed is not publicly available for ethical and data safety reasons. However, the same dataset can be constructed by request to the Swedish National Board of Health and Welfare and Statistics Sweden after approval of the research project by an Ethical Committee and by the data safety committees of the Swedish Authorities. The study also needs to be performed in collaboration with Swedish researchers. [1] 1. Public Access to Information Secrecy Act. In: Justice Mo, editor. Stockholm2009.

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
