# Peer review of "Carpal Tunnel Syndrome and Ulnar Nerve Entrapment Are Associated with Impaired Psychological Health in Adults as Appraised by Their Increased Use of Psychotropic Medication"

_jcm, 2022, doi:10.3390/jcm11133871_

Round 1

Reviewer 1 Report

Dear Authors: I want to congratulate with You for Your work. the search is incredibly wide, and cross-linked studies from different databases such this one are interesting. of course, biases can be related to a variety of different factors that could not be evaluated; this is a limit of bigdata studies. best regards.

Author Response

Dear Authors: I want to congratulate with You for Your work. the search is incredibly wide, and cross-linked studies from different databases such this one are interesting. of course, biases can be related to a variety of different factors that could not be evaluated; this is a limit of bigdata studies. best regards.

Thank you very much for your positive judgement

Reviewer 2 Report

The manuscript: “Carpal tunnel syndrome and ulnar nerve entrapment are associated with impaired psychological health in adults as appraised by their increased use of psychotropic medication” is prepared on the basis of several Swedish national registers that cover entire country population (almost 9.5 millions and 5.751 after exclusions). Research conducted on such large databases gives reliable results. In the population assessed in 2010, there were 9,728 patients with CTS, and 2,732 people treated with psychotropic drugs. Data on diagnoses and medications taken are from 2011-2012. Thus, there are differences in years between databases, which may require a brief comment from the authors:

·      In line 86 the authors state: “Our research database consisted of the total Swedish population of 2010”.

·      In lines 110-111 the authors state: “We assigned an individual baseline date to every individual, defined by the date of the first CTS/UNE diagnosis in 2011, or 1st January 2011 if the individual did not have any CTS/UNE diagnosis”.

So, was the untreated population from 2010 and the data on treated patients from 2011-2012?

Some comments also apply to the Discussion chapter:

·      In lines 241-244 authors state: “In the unadjusted analysis, both absolute and relative risk of psychotropic medication use was much higher in individuals that were diagnosed and surgically treated for CTS and UNE or both conditions; a combination that is common in clinical practice.

However, in Table 1 authors stated that this combination is low (0,00%). Please formulate the sentence in lines 241-244 differently as it seems a bit inaccurate.

In turn, the discussion lacks attempts to refer to the possible common etiology of impaired mental health and CTS. Can poor socio-economic conditions be one of the common causes? 

·      In lines 258-259 the authors refer to an association between low economic well-being and CTS. 

·      In lines 267-269, they refer to the association between low income, low occupational level and psychotropic drug use. 

·      In turn, in lines 276-277 they raise an interesting problem of the possibility of improving depressive symptoms after surgery. However, this sentence is included in the paragraph that concerns the risk of psychotropic drugs use with UNE and with CTS. 

Perhaps combining these considerations in a separate paragraph referring to the possible primary etiological factors of CTS, UNE and depression could enrich the discussion.

Author Response

The manuscript: “Carpal tunnel syndrome and ulnar nerve entrapment are associated with impaired psychological health in adults as appraised by their increased use of psychotropic medication” is prepared on the basis of several Swedish national registers that cover entire country population (almost 9.5 millions and 5.751 after exclusions). Research conducted on such large databases gives reliable results. In the population assessed in 2010, there were 9,728 patients with CTS, and 2,732 people treated with psychotropic drugs. Data on diagnoses and medications taken are from 2011-2012. Thus, there are differences in years between databases, which may require a brief comment from the authors:

  • Question:

 In line 86 the authors state: “Our research database consisted of the total Swedish population of 2010”.

  • In lines 110-111 the authors state: “We assigned an individual baseline date to every individual, defined by the date of the first CTS/UNE diagnosis in 2011, or 1st January 2011 if the individual did not have any CTS/UNE diagnosis”.

So, was the untreated population from 2010 and the data on treated patients from 2011-2012?

Reply:

Thank you very much for your positive assessment. Our population consisted of people living in Sweden December 31st, 2010. Then, we studied the use of psychotropics from January 1st, 2011 until December 31st, 2012. We investigated the use of psychotropics during the first year after a diagnosis 2011 and those with no diagnoses were followed from the January 1st2011.

For instance, an individual with CTS diagnosis June 15th 2011 was followed until June 15th 2012, and one individual with CTS December 31 st 2011 was followed until December 31 st 2012. Those individuals without any CTS diagnoses were followed for the whole year 2011.

Some comments also apply to the Discussion chapter:

  • Question:

In lines 241-244 authors state: “In the unadjusted analysis, both absolute and relative risk of psychotropic medication use was much higher in individuals that were diagnosed and surgically treated for CTS and UNE or both conditions; a combination that is common in clinical practice.

However, in Table 1 authors stated that this combination is low (0,00%). Please formulate the sentence in lines 241-244 differently as it seems a bit inaccurate.

Reply:

Yes, we agree that this expression could create some confusion. We have deleted the last part of the sentence and rephrased it in another way in a separate sentence. Thus, from a clinical perspective the combination of the two nerve compression syndromes simultaneously is noticeable, while from a population perspective the prevalence of this combined syndrome is very low. Changes have been made in Discussion. 

Question:

In turn, the discussion lacks attempts to refer to the possible common etiology of impaired mental health and CTS. Can poor socio-economic conditions be one of the common causes? 

Reply:

Yes, poor socioeconomic status could be a confounder. The socioeconomic status is a common cause of both the nerve compression disorders and impaired psychological health. However, the adjusted analyses (Table 2, model 2) indicated an independent effect of both the disorders and socioeconomic status.  This is clarified in Discussion.

  • Question

In lines 258-259 the authors refer to an association between low economic well-being and CTS. 

  • In lines 267-269, they refer to the association between low income, low occupational level and psychotropic drug use. 
  • In turn, in lines 276-277 they raise an interesting problem of the possibility of improving depressive symptoms after surgery. However, this sentence is included in the paragraph that concerns the risk of psychotropic drugs use with UNE and with CTS. 

Perhaps combining these considerations in a separate paragraph referring to the possible primary etiological factors of CTS, UNE and depression could enrich the discussion.

Reply
We agree that it is a complexity of the relation between nerve compression disorders, such as CTS, mental health and socioeconomic status, where also the aetiology of the nerve compression disorder may be multifactorial. We have added a sentence in Discussion raising this issue.

Reviewer 3 Report

Overall it's great job. The correlation of psychotropic drugs and neuropathies is known, but the association with work and social and health conditions in the various age groups is interesting and useful. Very interesting, as can be seen from the line chart in figure 2, the lines of men and women without neuropathy grow slowly compare to the lines with both neuropathies which fluctuate  respect to the percentage of psychotropic drugs. The most frequent  is ulnar neuropathy in middle-aged women in relation to high percentages of drugs. The conclusions are unclear when it is best to avoid surgery in these patients. I would only elaborate on this final part.

Author Response

Overall it's great job. The correlation of psychotropic drugs and neuropathies is known, but the association with work and social and health conditions in the various age groups is interesting and useful. Very interesting, as can be seen from the line chart in figure 2, the lines of men and women without neuropathy grow slowly compare to the lines with both neuropathies which fluctuate  respect to the percentage of psychotropic drugs. The most frequent  is ulnar neuropathy in middle-aged women in relation to high percentages of drugs. The conclusions are unclear when it is best to avoid surgery in these patients. I would only elaborate on this final part.

 Reply
Thank you for these comments. We have clarified this idea in the revised manuscript.